# 7T HR FID-MRSI Compared to Amino Acid PET: Glutamine and Glycine as Promising Biomarkers in Brain Tumors

**DOI:** 10.3390/cancers14092163

**Published:** 2022-04-26

**Authors:** Gilbert Hangel, Philipp Lazen, Sukrit Sharma, Barbara Hristoska, Cornelius Cadrien, Julia Furtner, Ivo Rausch, Alexandra Lipka, Eva Niess, Lukas Hingerl, Stanislav Motyka, Stephan Gruber, Bernhard Strasser, Barbara Kiesel, Matthias Preusser, Thomas Roetzer-Pejrimovsky, Adelheid Wöhrer, Wolfgang Bogner, Georg Widhalm, Karl Rössler, Tatjana Traub-Weidinger, Siegfried Trattnig

**Affiliations:** 1High-Field MR Center, Department of Biomedical Imaging and Image-Guided Therapy, Medical University of Vienna, 1090 Vienna, Austria; philipp.lazen@meduniwien.ac.at (P.L.); e1429163@student.tuwien.ac.at (S.S.); barbara.hristoska@stud.fh-campuswien.ac.at (B.H.); cornelius.cadrien@meduniwien.ac.at (C.C.); alexandra.lipka@meduniwien.ac.at (A.L.); eva.niess@meduniwien.ac.at (E.N.); lukas.hingerl@meduniwien.ac.at (L.H.); stanislav.motyka@meduniwien.ac.at (S.M.); stephan.gruber@meduniwien.ac.at (S.G.); bernhard.strasser@meduniwien.ac.at (B.S.); wolfgang.bogner@meduniwien.ac.at (W.B.); siegfried.trattnig@meduniwien.ac.at (S.T.); 2Department of Neurosurgery, Medical University of Vienna, 1090 Vienna, Austria; barbara.kiesel@meduniwien.ac.at (B.K.); georg.widhalm@meduniwien.ac.at (G.W.); karl.roessler@meduniwien.ac.at (K.R.); 3Department Applied Life Sciences, University of Applied Sciences Campus Vienna, 1100 Vienna, Austria; 4Division of Neuroradiology and Musculoskeletal Radiology, Department of Biomedical Imaging and Image-Guided Therapy, Medical University of Vienna, 1090 Vienna, Austria; julia.furtner@meduniwien.ac.at; 5Center for Medical Physics and Biomedical Engineering, Medical University of Vienna, 1090 Vienna, Austria; ivo.rausch@meduniwien.ac.at; 6Division of Oncology, Department of Internal Medicine I, Medical University of Vienna, 1090 Vienna, Austria; matthias.preusser@meduniwien.ac.at; 7Division of Neuropathology and Neurochemistry, Department of Neurology, Medical University of Vienna, 1090 Vienna, Austria; thomas.roetzer@meduniwien.ac.at (T.R.-P.); adelheid.woehrer@meduniwien.ac.at (A.W.); 8Division of Nuclear Medicine, Department of Biomedical Imaging and Image-Guided Therapy, Medical University of Vienna, 1090 Vienna, Austria; tatjana.traub-weidinger@meduniwien.ac.at; 9Institute for Clinical Molecular MRI, Karl Landsteiner Society, 3100 St. Pölten, Austria; 10Christian Doppler Laboratory for Clinical Molecular MR Imaging, Christian Doppler Research Association, 1090 Vienna, Austria

**Keywords:** 7T, MRSI, PET, gliomas, MR spectroscopy, glutamine, glycine, choline

## Abstract

**Simple Summary:**

Magnetic resonance spectroscopic imaging is an imaging method that can map the distribution of multiple biochemicals in the human brain in one scan. Using stronger magnetic fields, such as 7 Tesla, allows for higher resolution images and more biochemical maps. To test these results, we compared it to positron emission tomography, the established clinical standard for metabolic imaging. This comparison mainly looked at the overlap between regions with increased signal between both methods. We found that the molecules glutamine and glycine, only mappable at 7 Tesla, corresponded better to positron emission tomography than the commonly used choline.

**Abstract:**

(1) Background: Recent developments in 7T magnetic resonance spectroscopic imaging (MRSI) made the acquisition of high-resolution metabolic images in clinically feasible measurement times possible. The amino acids glutamine (Gln) and glycine (Gly) were identified as potential neuro-oncological markers of importance. For the first time, we compared 7T MRSI to amino acid PET in a cohort of glioma patients. (2) Methods: In 24 patients, we co-registered 7T MRSI and routine PET and compared hotspot volumes of interest (VOI). We evaluated dice similarity coefficients (DSC), volume, center of intensity distance (CoI), median and threshold values for VOIs of PET and ratios of total choline (tCho), Gln, Gly, myo-inositol (Ins) to total N-acetylaspartate (tNAA) or total creatine (tCr). (3) Results: We found that Gln and Gly ratios generally resulted in a higher correspondence to PET than tCho. Using cutoffs of 1.6-times median values of a control region, DSCs to PET were 0.53 ± 0.36 for tCho/tNAA, 0.66 ± 0.40 for Gln/tNAA, 0.57 ± 0.36 for Gly/tNAA, and 0.38 ± 0.31 for Ins/tNAA. (4) Conclusions: Our 7T MRSI data corresponded better to PET than previous studies at lower fields. Our results for Gln and Gly highlight the importance of future research (e.g., using Gln PET tracers) into the role of both amino acids.

## 1. Introduction

In vivo magnetic resonance spectroscopic imaging (MRSI) allows for imaging of neurochemical compounds in the human brain. MRSI benefits particularly from ultra-high-field (UHF) MRI, e.g., 7 Tesla (7T) [1,2,3,4,5], due to increased signal-to-noise ratio (SNR) and improved spectral separation, but there are also major technical challenges. Most of these can be addressed by free-induction-decay (FID) MRSI techniques [2,6]. Recently, we have introduced a 7T MRSI method using spatial–spectral encoding that scans the human cerebrum with a nominal isotropic resolution of 3.4 mm within 15 min [7]. With this, 7T MRSI has potentially overcome the limitations regarding scan time, resolution, coverage, and molecules of interest that held back clinical routine MRSI.

The first clinical FID-MRSI studies yielded promising results for neuro-oncology. Beyond higher spatial resolutions, the most promising feature is the better separation of oncometabolites such as glutamine (Gln) from glutamate (Glu) [8,9,10,11] or glycine (Gly) from Ins [7,9]. The precise role of the amino acids Gln and Gly in tumors is not well investigated, but Gln has emerged as a central precursor in the metabolism of cancer cells [12] and Gly is linked to cell proliferation [13]. Both were found to increase heterogeneously within high-grade gliomas [9], and altered levels of both have been connected to antigen expression in cancer stem cells [14]. In contrast, clinical routine MRSI at ≤3T is generally limited to the quantification of three metabolites: total choline (tCho); total creatine (tCr); and total N-acetyl-aspartate (tNAA). Other metabolites require special methods such as spectral editing [15] for the detection of 2-hydroxyglutarate [16].

Positron emission tomography (PET) is a well-accepted technique for imaging neuro-oncometabolism. It features a similar resolution to MRSI and an array of possible tracers with which to image different metabolic pathways. Due to the natural high glucose uptake in the brain, 2-[F-18]fluoro-2-deoxy-D-glucose(FDG)-PET [17] has been largely replaced by radiolabeled amino acids, such as [S-methyl-11C]methionine (MET)-PET [18], and O-(2-[F-18]fluoroethyl)-L-tyrosine(FET)-PET [19]. These are able to visualize tumoral activity [17,20] as surrogates for amino acid transporter activity in cell membranes, which is linked to tumor metabolism and growth. Their specific clinical applications as biomarkers include the prognosis of tumor malignancy using FET-PET [21], where higher relative uptake is a negative predictor for patient survival in high-grade gliomas. Both FET- and MET-PET can be employed to differentiate between high- and low-grade gliomas based on relative uptake as well [22,23].

As we further investigate the performance and robustness of our method, the verification of the extent of MRSI-observed metabolic changes in tumors by an established method is necessary. As a metabolic imaging technique with a strong clinical presence, PET is the logical choice, as the clinically available methods of PET perform better than these of MRS [24]. The main onco-metabolite imaged with MRSI, tCho, represents membrane turnover [25] and is only indirectly related to amino acids. Nevertheless, a significant correlation of tCho to MET-PET and tCho/tNAA to FET-PET has been found [26,27,28]. In the most comprehensive studies comparing FET-PET and 3D-MRSI in 41 glioma patients [29], the FET tCho/tNAA hotspots overlapped by 40 ± 25%. Investigating IDH mutations, studies successfully used FET PET-guided SVS to determine IDH status in gliomas, identifying Gly and Ins as important biomarkers [30] or used simultaneous PET and SVS to the same end, finding equal performance for SVS and PET [31]. To the best of our knowledge, no prior research has explored the spatial correlation of amino-acid PET and 7T MRSI, with the capability to image the amino acids Gln/Gly. Although MRSI does not image the exact same amino acids and their processes, a much higher correspondence to FET and MET uptake can be expected for Gln/Gly compared to tCho, which is only indirectly related to amino acid uptake.

### Purpose

To investigate whether the metabolic changes found by our 7T MRSI method correspond to the established metabolic imaging method PET by correlating the multiple neuro-oncological markers (tCho, Gln, Gly, Ins ratios to tNAA and tCr) of 7T MRSI to clinical routine FET- and MET-PET in a cohort of glioma patients.

## 2. Materials and Methods

### 2.1. Subject Recruitment

From 37 glioma patients who participated in 7T-MRSI [9] between October 2018 and April 2021, we retrospectively included 28 patients (mean age 51 years, range 26–77 years, 10 female) who had received routine PET scans within two months (20 in the same month) of the 7T protocol prior to surgery (15 with FET and nine with MET). IRB approval and written, informed consent were obtained. All subjects were screened for 7T MRI contraindications (e.g., pregnancy, claustrophobia, ferromagnetic implants, non-ferromagnetic metal head implants > 12 mm) and a Karnofsky performance status > 70 prior to the 7T scan. The histology of the tumors was retrospectively updated to the 2021 WHO classification [32]. Four patient scans were excluded due to low MRSI quality. The remaining 24 patients are described in Table 1, while recruitment is illustrated in Appendix A. Ten high-grade glioma cases were previously reported qualitatively in [9].

### 2.2. 7T MRSI Measurement and Processing

We used a 7T scanner (Siemens Healthineers, Erlangen, Germany, Magnetom/Magnetom plus after an upgrade) and a 32-channel head receive coil array (Nova Medical, Wilmington, MA, USA). Prior to MRSI, we measured MP2RAGE as T1w-MRI and FLAIR as T2w-MRI. Parameters for these and MRSI are listed in Appendix A. The MRSI sequence itself [7] used a free-induction-decay acquisition, concentric circle trajectories with a FOV of 220 × 220 × 133 mm^3^, and resolution of 3.4 × 3.4 × 3.4 mm^3^. At a TR of 450 ms and an acquisition delay of 1.3 ms, MRSI acquisition took 15 min. We placed the FOV in parallel to the corpus callosum horns, aiming to cover as much of the morphologically visible tumor as possible with preference for the superior parts of the brain (due to B0-inhomogeneities more strongly affecting the basal regions). Further parameters were 39° excitation flip angle, 345 ms readout duration with 2778 Hz spectral bandwidth, and 7T-optimized WET water suppression [33], but no lipid suppression during acquisition to maintain a short TR.

We processed the MRSI data offline using a custom framework [34]. Processing featured an iMUSICAL coil combination [35,36] and k-space regridding [35]. Coil-wise L2-regularization [37] was performed to remove lipid signals. The resulting spectra were voxel-wise quantified using LCModel. A molecular basis of (tCr/γ-aminobutyric acid/Gln/Glu/Gly/glutathione/tCho/Ins/serine/taurine/NAA/NAA-glutamate/macromolecular baseline [38]) was fit in the 1.8–4.1 ppm range. The resulting intensities (in institutional units) were aggregated into 3D metabolic maps. For the evaluation of spectral quality, we calculated, voxel-wise, the full width at half maximum (FHWM) and the signal-to-noise ratio (SNR) of tCr at 3.02 ppm, as well as Cramér–Rao lower bounds (CRLB) of all metabolites [39]. For every metabolite map, voxels with at least one of tCr SNR < 5, tCr FWHM > 0.15 ppm, or metabolite CRLB > 40% were filtered out and excluded from further analysis. Our MRSI methods are described in greater detail in previous publications [7,9], and summarized according to the MRSinMRS standard in Appendix A [40].

In addition, we collected routine 3T MRI data [41], including at least T1-weighted imaging (T1w) with/without contrast-enhancement (Gadoteridol, 0.1 mmol/kg), T2-weighted imaging (T2w), and fluid-attenuated inversion recovery (FLAIR) from which a tumor segmentation (T-segmentation, only tumor yes/no) was derived by a neuroradiologist (J.F.) [9].

### 2.3. PET Measurement and Processing

PET imaging was achieved using Siemens Biograph mMR PET/MR and Biograph TPTV PET/CT systems (Siemens Healthineers, Erlangen, Germany). Radiotracer was administered intravenously as a bolus with target doses of 250MBq for FET and 700MBq for MET. PET imaging was performed either as a dynamic acquisition over 40 min for FET or as a static scan starting at 20 min post tracer injection for MET. Data used in this study were the sum image of the last 10 min of the dynamic acquisition (30–40 min post injection) for FET or a 10 min static acquisition for MET. Image reconstruction was accomplished using OSEM algorithms into a 172 × 172 or 168 × 168 matrix with a zoom factor of 2, resulting in a pixel size of approximately 2 × 2 mm. Attenuation correction was performed using the standard Dixon-based approach as described in [42]. A 3 or 5 mm FWHM Gaussian post reconstruction filter was applied to the images. We then calculated tumor-to-brain ratio (TBR) maps, dividing voxel activity values by the mean activity of a normal-appearing white matter (NAWM) control region, as defined below.

### 2.4. Co-Registration and Comparison

We co-registered clinical 3T MRI, 7T MRSI, and PET images using MITK (Medical Imaging Interaction Toolkit). We derived a white matter (WM) segmentation from 7T-T1w images, including only voxels with >90% WM content. Tumor and WM segmentation, as well as PET maps, were then resampled to the MRSI resolution stated above.

From the T-segmentation, we created a second segmentation, including a peritumoral region (P-segmentation), using six iterations of dilation (corresponding to 2 cm). This P-segmentation was then removed from the NAWM control segmentation, which was eroded once afterward.

As an estimation of the concentration of metabolites within the tumor was not possible without a knowledge of intratumoral T1 and water concentrations, we chose metabolite ratios instead. We decided to use tCho/tNAA and tCho/tCr as markers most comparable to previous literature [26,27,28,29], Gln/tNAA, Gln/tCr, and Gly/tNAA, Gly/tCr as 7T MRSI onco-markers of interest, together with Ins/tNAA and Ins/tCr as complements to Gly.

For the definition of VOI cutoff thresholds both for PET and MRSI necessary for quantitative evaluation, we used previous PET/MRSI studies as guidelines and to maximize comparability to our study. These studies used TBR cutoffs ranging from 1.15 for an unspecific uptake increase [26] up to 1.5/1.6 for pathological changes [19,26,29], or even 2.0 for hot lesions [28,43], while current standards for pathological uptake would be TBRs of >1.6 for FET and >1.3 for MET [44]. MRSI values in comparison studies were considered only within previously defined PET VOIs [27,28] or cutoffs defined as 2.0 times the mean value of tCho/tNAA in an NAWM control region [29,45]. Based on this, we arrived at three cutoff values applied both to the lower bound of values for MRSI and PET, namely, 1.15 for an unspecific metabolic increase, 1.6 for a clear oncometabolic change, and 2.0 for the activity core. In our NAWM segmentation, we calculated median values for all used metabolite ratios to calculate these cutoffs. While TBR calculation used mean values, we used medians for MRSI ratios as this is more robust to outliers. All three cutoffs were applied, respectively to all TBR and MRSI ratios, within the T-segmentation resulting in the VOIs “TSEG1.15”, ”TSEG1.6”, ”TSEG2.0”, and within the P-segmentation resulting in the VOIs “PSEG1.15”, ”PSEG1.6” and ”PSEG2.0”.

For these six VOIs, we calculated, patient-wise, the Sørensen–Dice similarity coefficient (DSC) between MRSI ratios and PET TBR:(1)$$DSC=\frac{2\times \left|N_{MRSI}\;\cap \;N_{PET}\right|}{\left|N_{MRSI}\right|+\left|N_{PET}\right|}$$
with $\left|N_{MRSI}\right|$ and $\left|N_{PET}\right|$ as MRSI and PET VOI voxel amounts. A DSC of 1 would describe a total overlap of two volumes, while a DSC of 0 would signify no common voxels. In addition, DSCs were also calculated between MRSI and T-segmentation, as well as PET TBR and T-segmentation. Two final DSCs were calculated between PET TBR and the summed VOIs of all respective ratios to tNAA and tCr (Sum/tNAA and Sum/tCr). We further calculated the distance between the center of intensity between PET and MRSI VOIs:(2)$$d_{P-M}=\left|{\vec{r}}_{P}-{\vec{r}}_{M}\right|$$
with the volume intensity vectors
(3)$${\vec{r}}_{VOI}=\frac{\sum_{i\in VOI}{\vec{v}}_{i}\times I\left({\vec{v}}_{i}\right)}{\sum_{i\in VOI}I\left({\vec{v}}_{i}\right)}$$
and ${\vec{v}}_{i}$, the individual spatial voxel vectors and their corresponding intensities $I\left({\vec{v}}_{i}\right)$, similar to [29]. VOI volumes, medians within the VOIs, and MRSI VOI thresholds were also recorded. We further evaluated correlations between these derived values, tumor grade, and IDH status within the study cohort. Our workflow is summarized in Appendix A.

## 3. Results

Our evaluation could be performed in 24 patients with sufficient MRSI quality. Example spectra are available in Appendix A. While this report focuses on the overall cohort results (e.g., boxplots for TSEG1.6 in Figure 1), detailed patient-level data are available in the Appendix A. Overall, a threshold of 1.15 proved to be unspecific for VOI definition, as expected from previous literature. This is demonstrated in histograms of the DSCs for Gln/tNAA for all six VOIs in Figure 2.

Both 1.6 and 2.0 thresholds resulted in better localized VOIs of PET and MRSI alike. At 1.6, two PET VOIs, and at 2.0, five PET VOI volumes were <1 cm^3^ within T-segmentation. An overview of PET TBR, MRSI ratio, and VOI maps for multiple patients (Figure 3) demonstrates inter-metabolite heterogeneity and different correspondences between VOI cutoffs. Figure 4 details a case with high correspondence between T-segmentation, MRSI, and PET ratios, with a cutoff of 1.6 and 2.0 for T-segmentation. In contrast, Figure 5 shows a patient with limited PET/MRSI correspondence and high heterogeneity between MRSI ratios. All PT-VOIs showed greatly increased volumes compared to the Seg-VOIs, specifically for PET, Gln/tNAA, and Gly/tNAA at 1.6 and 2.0 cutoffs. CoI distances increased heterogeneously and were more pronounced for 2.0 over 1.6.

### 3.1. DSC

Despite cases with minimal PET/MRSI VOI overlap, overall DSCs, as presented in Table 2, showed good comparability between PET and MRSI. DSCs decreased with increased cutoffs, for example, in Gln/tNAA from 0.82 ± 0.27 for TSEG1.15 to 0.66 ± 0.40 for TSEG1.6 and 0.49 ± 0.52 for TSEG2.0. Ratios to tNAA generally had higher DSCs than those to tCr, and Seg-VOIs lower than PT-VOIs. Over the whole cohort, Gln-ratios always resulted in higher DSCs than tCho ratios, with Gly performing between. As a further example of the high correspondence of Gln/tNAA to PET, for TSEG1.6, 16/24 datasets resulted in a DSC > 0.5. Ins/tNAA and ns/tCr showed the least correspondence. The VOI sums resulted in the highest DSCs, but were only slightly higher than Gln.

Compared to the T-segmentation, Gln showed higher DSCs than PET in the TSEG1.6 and TSEG2.0 cases and similar values for all P cases, as detailed in Table 3, with Gly, tCho, and Ins following. Only in the PSEG1.15 VOI did the larger sum VOIs result in reduced DSCs to the T-segmentation.

### 3.2. Distance

CoI distances between PET and MRSI ratios (Table 4) were the least for Gln/tNAA (except for PSEG1.15, with Gln/tCr), with 0.21 ± 0.30 cm for TSEG1.15, 0.39 ± 0.22 cm for TSEG1.6, and 0.63 ± 0.92 cm for TSEG2.0. All median ratios for the T-VOIs except Ins/tCr were below 1 cm. Ratios to tNAA had less CoI distance to PET than those to tCr, with Ins ratios having the highest median distance.

### 3.3. Volume

Median VOI volumes were greatest for Gln/tNAA and Gln/tCr (Table 5). Volumes were heterogeneous over the ratios with regard to cutoffs and masks. PET, Gln, and Gly volumes were greater than the T-segmentation in the PSEG1.6 case.

### 3.4. Medians and Thresholds

Medians (Table 6) increased with higher cutoff values but not at the same rate; for example, the PET TBR of 1.86 ± 0.53 for TSEG1.15 up to 2.42 ± 0.30 for TSEG2.0, or Gln/tNAA from 0.59 ± 0.31 for TSEG1.15 up to 0.68 ± 0.27 for TSEG2.0. This indicates that large parts of the VOIs were well above thresholds. Median ratios to tCr were consistently higher than those to tNAA.

Thresholds (Table 7) showed more variation than expected, such as 0.34 ± 0.16 median and IQR for Gln/tNAA. Still, VOI medians were clearly above the cutoff thresholds.

### 3.5. Correlations

We observed strong positive correlations between DSCs (except Ins/tNAA and Ins/tCr) and strong negative correlations between DSC and CoI distance. High correlation between volumes was observed for all six cases. Our data did not show any strong correlations between image parameters and low/high WHO grade except for a correlation in the −0.5 to −0.6 range for the DSC of Ins/tNAA to PET over all VOIs. IDH status had the strongest correlation of −0.53 to Ins/tCr in TSEG1.6. An example correlation matrix for TSEG1.6 is given as Figure 6, with all others being found in the Appendix A.

Finally, an evaluation grouped into tumor classification and grade for the TSEG1.6 case, presented in Appendix A, while limited by small group sizes, shows that DSCs are generally higher for high-grade tumors than for low-grades, with the exception of grade 3 oligodendrogliomas, which have the least correspondence. This is apparent for the comparison of MRSI to PET as well as MRSI to T-SEG.

## 4. Discussion

Our study compared 7T MRSI to clinical PET for the first time. We confirmed higher correspondence of the amino acids Gln and Gly to amino acid PET than the spectroscopic standard of tCho.

To the best of our knowledge, there are only four studies that aimed to compare 1.5T/3T MRSI to PET thus far. Two older publications by Stadlbauer et al. [27] and Widhalm et al. [26] conducted only topographical correlations, limiting comparability. The first one at 1.5T found a correlation of >50% in 11/15 patients (tCho/tNAA), but used regions defined by the percentage of PET maxima, while the second, at 3T, used a PET TBR cutoff of 1.5 and MRSI ratio cutoffs of 1.0, resulting in >50% correlation for 18/21 glioma patients (tCho/tNAA and tCho/tCr) with PET and MRSI hotspots. Our study’s TSEG1.6 case found DSCs > 50% for 13 (tCho/tNAA), 16 (Gln/tNAA), 15 (Gly/tNAA), and 5 (tCho/tCr) of 21 patients with PET uptake >1 cm^3^ in the T-segmentation. Despite these methodological differences, these results are broadly similar, except for tCho/tCr.

Bisdas et al. [28] compared MET with MRSI (tCho/tNAA and tCho/tCr) at 3T. With a TBR threshold of 2.0, in only three of 28 patients, MRSI and MET uptake overlapped >50%. Compared to that, our TSEG2.0 case had 10/24 DCSs > 0.5 for tCho/tNAA, showing a higher rate of correspondence. The most quantitative MRSI/PET comparison, to date, by Mauler et al. [29] included 41 glioma patients, and compared VOI volume, DSCs, and CoI distance using 3T. Using a TBR cutoff of 1.6 and a tCho/tNAA cutoff of 2/3, this study is most similar to our TSEG1.6 results. Over all patients, they found mean volumes of 19.12 ± 20.29 cm^3^ for FET and 21.79 ± 23.72 cm^3^ for tCho/tNAA, a DSC of 0.40 ± 0.25, and a distance of 0.93 ± 0.79 cm. In comparison, our study found 24.33 ± 30.46 cm^3^ for PET, 19.08 ± 23.10 cm^3^ for tCho/tNAA, a DSC of 0.53 ± 0.36, a distance of 0.54 ± 0.43 cm, and even higher DSCs and volumes for Gln/tNAA and Gly/tNAA. Our data show a better PET/MRSI correspondence and similar volumes, despite different MRSI acquisitions. Our results accordingly show that more quantitative evaluations using well-defined thresholds allow for a better comparison of the MRSI studies. In summary, we found that our results are comparable to previous research but conform better to amino acid PET for tCho/tNAA, and that Gln and Gly have a better correspondence than tCho. Comparability to the literature is limited by heterogeneous MRSI and evaluation methods. Our results demonstrate the difficulty of defining VOI thresholds that remain comparable between subjects. As these directly affect VOI volumes, which will be relevant in further studies investigating the performance of MRSI in peritumoral regions, more data and standardization will be required.

Our results show that 7T MRSI can image the increased amino acid metabolism of gliomas. While our findings are similar to FET and MET PET, we have compared different amino acids and therefore different metabolic pathways. A more direct comparison, which requires the investigation of identical amino acids, could be made by adapting other PET tracers. 18F-fluoroglutamine [46] used to detect glutamine uptake has been demonstrated in humans and would be the ideal candidate to verify MRSI-based Gln mapping. Another possible investigation based on mapping Gln and Gly could target glioblastoma stem cells, which have been connected to tumor growth [14,47] and treatment resistance [48], as changes in Gly and Gln levels after stem cell injection were demonstrated using MRS in animal models [49]. In vitro research has identified higher Gln in cancer stem cells with CD44(+) expression compared to CD44(−), with Gln additionally related to glutamate and Notch signaling via glutaminase, while CD133(+) appears to be linked to Gly [14]. Connecting the presence of stem cell molecules to Gln/Gly during treatment could yield better understanding of the role of cancer stem cells and to their potential use for treatment monitoring.

### Limitations

A main limitation of our study is the small number of patients imaged with 7T MRSI and PET. Moreover, two different amino acid PET tracers were used, which was not ideal, but both reflect the amino acid transporter activity at the cell surface of glioma cells. Therefore, this first attempt to bring together 7T MRSI and amino acid PET-based data in glioma patients is acceptable.

Relying on ratios to tNAA or tCr introduces another factor of variation, but concentration estimation in brain tumors would require a robust intratumoral quantification of water. Previous challenges for MRSI methods [9], such as B0-/B1-inhomogeneity, as well as subject motion, still remain.

## 5. Conclusions

We, for the first time, compared 7T MRSI to routine PET to attempt to verify MRSI-observed metabolic changes with this established metabolic imaging method. Gln and Gly ratios showed a higher correspondence than the routinely used tCho ratios when compared to the established amino acid PET for glioma description. Larger study cohorts are needed to confirm our observations. In addition, more MRSI-specific PET tracers, such as 18F-fluoroglutamine, should be investigated to confirm our 7T method analysis. Further quantitative and standardized research can better define and understand the role of Gln/Gly in the pathogenesis and imaging of gliomas.

## Figures and Tables

**Figure 1 cancers-14-02163-f001:**
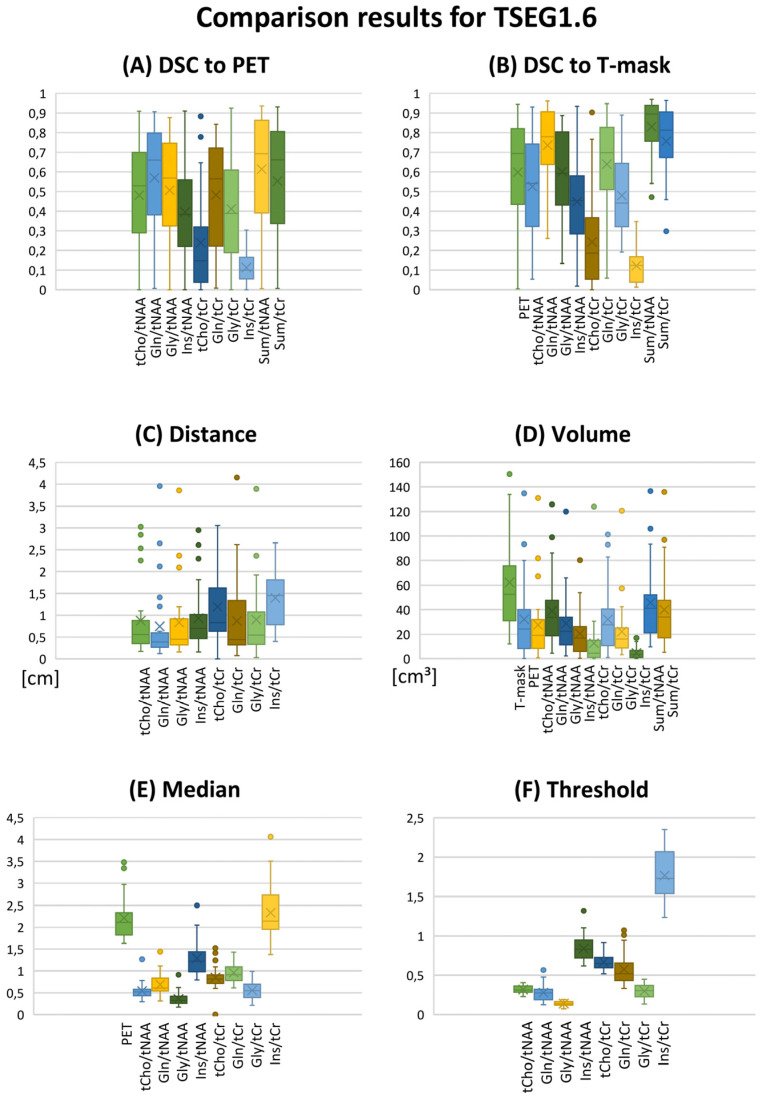
Overview of evaluation results as boxplots for the TSEG1.6 VOI definition. (**A**,**B**) DSCs for the comparison of MRSI ratios to PET and the T-segmentation. (**C**) CoI distance and (**D**) VOI volume. (**E**) Median ratios and (**F**) the calculated VOI thresholds. Gln/tNAA ratios show the best correspondence to PET. Key to the plot: Cross, mean; line, median; box, 2nd–3rd quartiles; whiskers, 1st and 4th quartiles.

**Figure 2 cancers-14-02163-f002:**
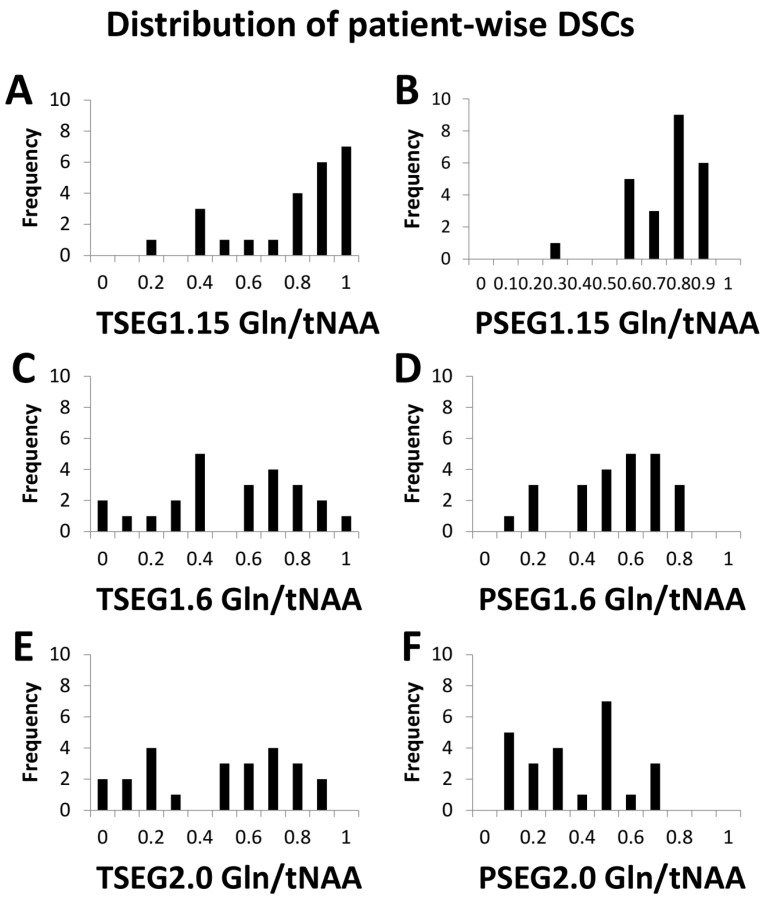
Histograms of the DCSs between PET TBR and Gln/tNAA for all six VOI definitions add to the finding that the TSEG1.15 and PSEG1.15 cases (**A**,**B**) were not as specific as the other two thresholds. The difference between 1.6 (**C**,**D**) and 2.0 (**E**,**F**) as cutoff appeared smaller.

**Figure 3 cancers-14-02163-f003:**
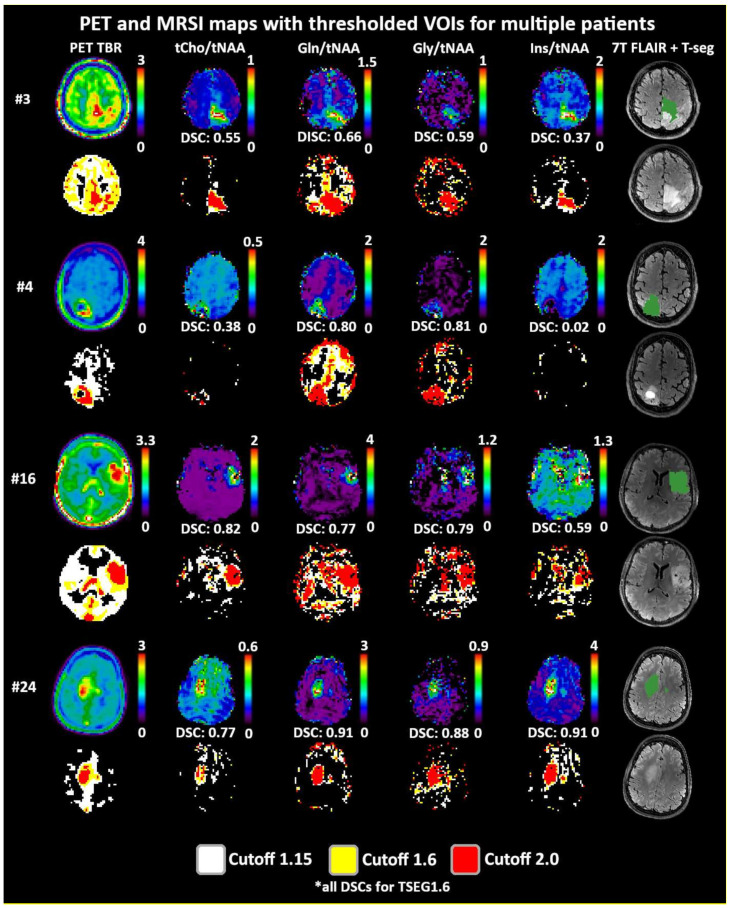
Overview of metabolic ratio images, PET TBR images, and the different applied thresholds, together with tumor segmentation for four patients. Heterogeneity between the different ratio maps is quite visible, as are the differences between cutoff values. The actual evaluation as performed only within the defined tumor segmentations is shown in green. PET maps were resampled to MRSI resolution.

**Figure 4 cancers-14-02163-f004:**
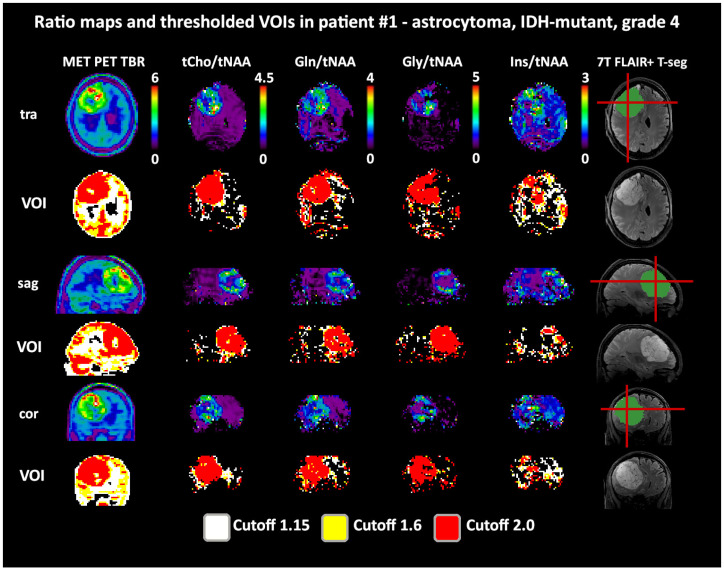
Example images of a patient with a high intratumoral correspondence between PET and MRSI ratios to tNAA (DSCs for TSEG1.6: 0.91/0.89/0.88/0.54 for tCho/Gln/Gly/Ins). Notably, the difference between the 1.6 and 2.0 cutoffs for the MRSI ratios is minimal. The actual evaluation as performed only within the defined tumor segmentations is shown in green. PET maps were resampled to MRSI resolution. Red lines indicate slice positions.

**Figure 5 cancers-14-02163-f005:**
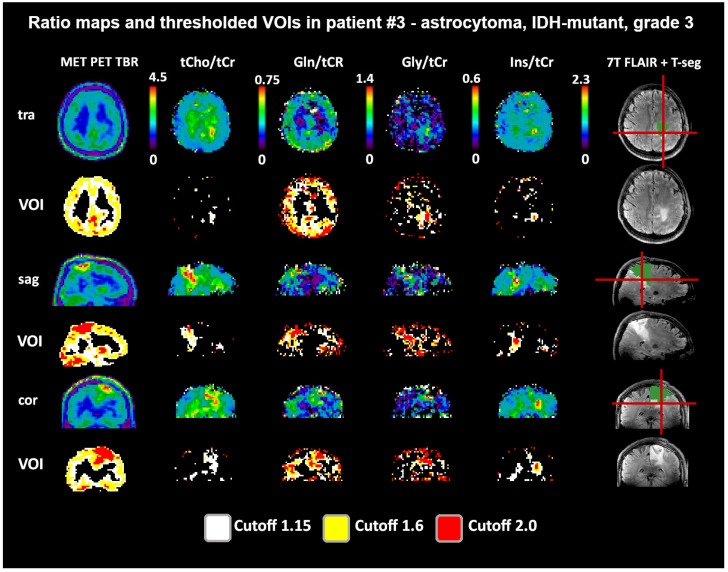
Example images of a patient with a low-to-moderate intratumoral correspondence between PET and MRSI ratios to tCr. While Gln/tCr and Gly/tCr align most directly with MET (DSCs for TSEG1.6: 0.59 and 0.55), tCho/tCr extends clearly beyond (DSC for TSEG1.6: 0.25), and Ins/tCr is located only more basally (DSC for TSEG1.6: 0.13). The actual evaluation as performed only within the defined tumor segmentations is shown in green. PET maps were resampled to MRSI resolution.

**Figure 6 cancers-14-02163-f006:**
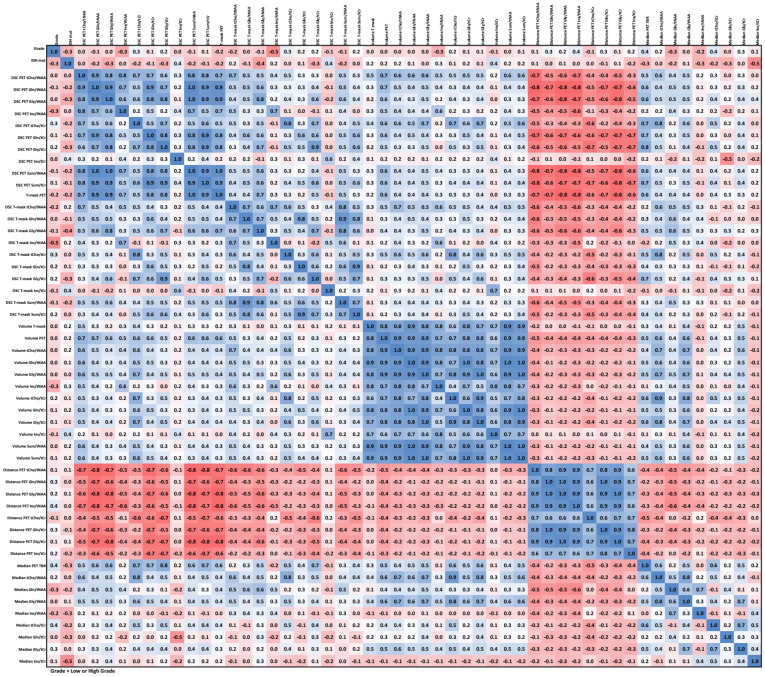
Correlation matrix for the TSEG1.6 VOI. An expected negative correlation between distance and the other parameters is the most visible feature. All correlation matrices are presented with higher readability in the Appendix A.

**Table 1 cancers-14-02163-t001:** Patient cohort overview.

Patient	Classification [WHO 2021]	Age	Sex	PET	IDH	TERT	MGMT Methylation	1p/19q Codeletion	CDKN2A/B hom.loss
1	Astrocytoma grade 4	51	male	MET	IDH1	C250T	yes	no	no
2	Astrocytoma grade 3	46	female	MET	IDH1	no	yes	no	no
3	Astrocytoma grade 3	29	male	MET	IDH1	no	yes	no	no
4	Glioblastoma grade 4	52	male	FET	WT	no	yes	N/A	N/A
5	Astrocytoma grade 2	33	male	FET	IDH1	no	yes	no	no
6	Astrocytoma grade 2	77	female	FET	IDH1	no	yes	no	no
7	Glioblastoma grade 4	65	female	MET	WT	C228T	no	N/A	N/A
8	Oligodendroglioma grade 3	51	male	FET	IDH1	C228T	yes	yes	no
9	Astrocytoma grade 3	62	male	MET	IDH1	no	yes	no	no
10	Diffuse hemispheric glioma grade 4	30	female	FET	WT	no	yes	no	yes
11	Astrocytoma grade 2	34	male	MET	IDH1	no	yes	no	no
12	Oligodendroglioma grade 3	56	male	FET	IDH1	N/A	yes	yes	yes
13	Astrocytoma grade 3	28	female	MET	IDH1	no	yes	no	no
14	Oligodendroglioma grade 2	50	female	MET	IDH1	no	yes	yes	no
15	Oligodendroglioma grade 2	38	female	FET	IDH1	no	yes	yes	no
16	Oligodendroglioma grade 2	61	male	FET	IDH1	C250T	yes	yes	no
17	Astrocytoma grade 2	33	male	MET	IDH1	no	no	no	no
18	Glioblastoma grade 4	58	male	FET	WT	N/A	no	N/A	N/A
19	Oligodendroglioma grade 3	57	female	FET	IDH1	N/A	yes	yes	N/A
20	Astrocytoma grade 3	40	male	FET	IDH1	no	yes	no	no
21	Glioblastoma grade 4	58	male	FET	WT	N/A	N/A	N/A	N/A
22	Astrocytoma grade 4	26	female	FET	IDH1	no	yes	no	yes
23	Glioblastoma grade 4	59	male	FET	WT	N/A	yes	N/A	N/A
24	Glioblastoma grade 4	46	female	FET	WT	no	no	no	yes

**Table 2 cancers-14-02163-t002:** DSCs to PET over all ratios and VOI thresholds.

DSC to PET	tCho/tNAA	Gln/tNAA	Gly/tNAA	Ins/tNAA	Sum/tNAA	tCho/tCr	Gln/tCr	Gly/TCr	Ins/tCr	Sum/tCr
TSEG1.15	0.71 ± 0.40	0.82 ± 0.27	0.67 ± 0.28	0.67 ± 0.39	0.90 ± 0.17	0.42 ± 0.34	0.80 ± 0.34	0.58 ± 0.23	0.33 ± 0.22	0.90 ± 0.27
TSEG1.6	0.53 ± 0.36	0.66 ± 0.40	0.57 ± 0.36	0.38 ± 0.31	0.69 ± 0.43	0.15 ± 0.24	0.56 ± 0.40	0.39 ± 0.38	0.10 ± 0.09	0.66 ± 0.42
TSEG2.0	0.36 ± 0.54	0.49 ± 0.52	0.43 ± 0.52	0.28 ± 0.33	0.49 ± 0.56	0.09 ± 0.20	0.38 ± 0.47	0.27 ± 0.48	0.05 ± 0.06	0.43 ± 0.50
PSEG1.15	0.51 ± 0.30	0.73 ± 0.19	0.52 ± 0.18	0.55 ± 0.20	0.80 ± 0.12	0.25 ± 0.15	0.71 ± 0.17	0.44 ± 0.17	0.26 ± 0.20	0.80 ± 0.17
PSEG1.6	0.32 ± 0.27	0.52 ± 0.31	0.39 ± 0.23	0.31 ± 0.26	0.56 ± 0.29	0.12 ± 0.09	0.45 ± 0.32	0.29 ± 0.20	0.10 ± 0.08	0.51 ± 0.33
PSEG2.0	0.25 ± 0.35	0.30 ± 0.34	0.31 ± 0.34	0.19 ± 0.24	0.30 ± 0.34	0.07 ± 0.09	0.26 ± 0.34	0.20 ± 0.28	0.04 ± 0.04	0.30 ± 0.34
	Medians and IQRs over all patients									

**Table 3 cancers-14-02163-t003:** DSCs to the T-Segmentation over all ratios and VOI thresholds.

DSC to T-Mask	PET TBR	tCho/tNAA	Gln/tNAA	Gly/tNAA	Ins/tNAA	Sum/tNAA	tCho/tCr	Gln/tCr	Gly/tCr	Ins/tCr	Sum/tCr
TSEG1.15	0.90 ± 0.25	0.67 ± 0.30	0.85 ± 0.22	0.71 ± 0.30	0.67 ± 0.20	0.92 ± 0.11	0.38 ± 0.37	0.79 ± 0.18	0.59 ± 0.30	0.30 ± 0.21	0.92 ± 0.15
TSEG1.6	0.69 ± 0.35	0.54 ± 0.37	0.78 ± 0.26	0.59 ± 0.33	0.45 ± 0.27	0.90 ± 0.18	0.19 ± 0.25	0.70 ± 0.30	0.44 ± 0.30	0.12 ± 0.11	0.81 ± 0.22
TSEG2.0	0.46 ± 0.54	0.46 ± 0.37	0.74 ± 0.30	0.54 ± 0.31	0.32 ± 0.23	0.85 ± 0.22	0.09 ± 0.17	0.62 ± 0.35	0.37 ± 0.22	0.06 ± 0.06	0.71 ± 0.30
PSEG1.15	0.34 ± 0.18	0.42 ± 0.19	0.39 ± 0.19	0.39 ± 0.14	0.35 ± 0.17	0.38 ± 0.14	0.29 ± 0.27	0.39 ± 0.19	0.35 ± 0.14	0.23 ± 0.14	0.38 ± 0.14
PSEG1.6	0.42 ± 0.31	0.41 ± 0.27	0.41 ± 0.20	0.38 ± 0.18	0.33 ± 0.20	0.42 ± 0.17	0.17 ± 0.24	0.35 ± 0.21	0.31 ± 0.18	0.10 ± 0.09	0.39 ± 0.18
PSEG2.0	0.39 ± 0.43	0.40 ± 0.30	0.42 ± 0.24	0.38 ± 0.22	0.26 ± 0.20	0.42 ± 0.19	0.08 ± 0.16	0.34 ± 0.24	0.26 ± 0.19	0.05 ± 0.05	0.37 ± 0.18
	Medians and IQRs over all patients										

**Table 4 cancers-14-02163-t004:** CoI distances of MRSI to PET over all ratios and VOI thresholds.

CoI Distance [cm]	tCho/tNAA	Gln/tNAA	Gly/tNAA	Ins/tNAA	tCho/tCr	Gln/tCr	Gly/tCr	Ins/tCr
TSEG1.15	0.36 ± 0.32	0.21 ± 0.30	0.34 ± 0.26	0.34 ± 0.36	0.63 ± 0.46	0.24 ± 0.36	0.33 ± 0.29	0.81 ± 0.43
TSEG1.6	0.56 ± 0.43	0.39 ± 0.22	0.45 ± 0.48	0.70 ± 0.46	0.84 ± 0.91	0.44 ± 0.94	0.55 ± 0.67	1.46 ± 0.97
TSEG2.0	0.71 ± 0.68	0.63 ± 0.92	0.61 ± 1.04	0.97 ± 0.85	0.93 ± 0.96	0.71 ± 0.90	0.70 ± 1.31	1.43 ± 1.14
PSEG1.15	0.52 ± 0.45	0.33 ± 0.25	0.32 ± 0.24	0.48 ± 0.50	0.73 ± 0.72	0.28 ± 0.31	0.41 ± 0.30	0.67 ± 0.52
PSEG1.6	0.65 ± 0.98	0.54 ± 0.76	0.57 ± 0.68	0.83 ± 0.88	1.24 ± 1.05	0.59 ± 0.74	0.81 ± 1.18	1.42 ± 1.06
PSEG2.0	1.01 ± 1.33	0.97 ± 0.92	1.11 ± 1.27	1.17 ± 1.12	1.55 ± 1.48	1.07 ± 0.80	1.32 ± 1.30	1.74 ± 1.42
	Medians and IQRs over all patients							

**Table 5 cancers-14-02163-t005:** VOI volumes for all ratios and VOI thresholds.

Volume [cm^3^]	T-Mask	PET	tCho/tNAA	Gln/tNAA	Gly/tNAA	Ins/tNAA	Sum/tNAA	tCho/tCr	Gln/tCr	Gly/TCr	Ins/tCr	Sum/tCr
**TSEG1.15**	52.47 ± 41.99	33.57 ± 40.25	29.67 ± 33.45	35.81 ± 24.91	23.96 ± 22.64	28.32 ± 17.89	43.90 ± 29.00	11.93 ± 23.03	30.78 ± 27.01	21.99 ± 22.15	11.24 ± 17.01	43.47 ± 29.96
**TSEG1.6**	52.47 ± 41.99	24.33 ± 30.46	19.08 ± 23.10	33.68 ± 24.60	22.38 ± 18.49	16.90 ± 15.86	41.09 ± 27.49	4.52 ± 12.36	27.63 ± 26.92	16.13 ± 14.06	3.44 ± 5.46	33.90 ± 28.71
**TSEG2.0**	52.47 ± 41.99	13.28 ± 23.96	14.01 ± 20.69	25.59 ± 23.22	21.07 ± 17.00	11.32 ± 12.70	35.59 ± 22.03	1.89 ± 6.63	22.89 ± 24.26	13.30 ± 11.21	1.75 ± 1.93	26.94 ± 20.85
**PSEG1.15**	52.47 ± 41.99	180.44 ± 125.94	84.21 ± 62.99	132.53 ± 84.20	83.40 ± 54.22	88.38 ± 53.65	192.18 ± 127.82	39.17 ± 34.25	122.92 ± 62.23	77.65 ± 51.67	47.15 ± 29.39	166.14 ± 121.33
**PSEG1.6**	52.47 ± 41.99	80.06 ± 100.20	37.97 ± 42.01	102.68 ± 60.21	65.22 ± 46.68	42.88 ± 33.70	148.49 ± 87.73	11.91 ± 14.72	88.49 ± 61.38	51.02 ± 41.27	13.32 ± 12.00	118.15 ± 67.73
**PSEG2.0**	52.47 ± 41.99	26.10 ± 57.49	25.06 ± 31.31	77.43 ± 58.93	48.11 ± 41.48	25.86 ± 21.01	114.87 ± 68.43	6.98 ± 8.46	62.26 ± 52.46	36.45 ± 31.54	7.07 ± 5.92	84.29 ± 49.31
	Medians and IQRs over all patients											

**Table 6 cancers-14-02163-t006:** VOI medians for all ratios and VOI thresholds.

Medians	PET TBR	tCho/tNAA	Gln/tNAA	Gly/tNAA	Ins/tNAA	tCho/tCr	Gln/tCr	Gly/tCr	Ins/tCr
TSEG1.15	1.86 ± 0.53	0.45 ± 0.14	0.59 ± 0.31	0.27 ± 0.17	0.90 ± 0.36	0.60 ± 0.11	0.82 ± 0.28	0.41 ± 0.23	1.49 ± 0.37
TSEG1.6	2.11 ± 0.42	0.52 ± 0.13	0.61 ± 0.25	0.33 ± 0.15	1.22 ± 0.36	0.83 ± 0.17	0.91 ± 0.29	0.55 ± 0.27	2.14 ± 0.73
TSEG2.0	2.42 ± 0.30	0.63 ± 0.14	0.68 ± 0.27	0.36 ± 0.14	1.42 ± 0.50	1.17 ± 0.39	1.01 ± 0.26	0.66 ± 0.30	2.88 ± 0.95
PSEG1.15	1.59 ± 0.27	0.33 ± 0.07	0.40 ± 0.10	0.21 ± 0.10	0.79 ± 0.22	0.57 ± 0.25	0.69 ± 0.25	0.40 ± 0.18	1.45 ± 0.38
PSEG1.6	1.88 ± 0.27	0.47 ± 0.11	0.45 ± 0.13	0.26 ± 0.12	1.11 ± 0.31	0.85 ± 0.27	0.85 ± 0.26	0.54 ± 0.24	2.20 ± 0.71
PSEG2.0	2.29 ± 0.27	0.59 ± 0.14	0.54 ± 0.15	0.32 ± 0.13	1.42 ± 0.43	1.14 ± 0.43	1.06 ± 0.27	0.69 ± 0.28	3.04 ± 1.08
	Medians and IQRs over all patients								

**Table 7 cancers-14-02163-t007:** Explicit VOI thresholds for all ratios and VOI threshold definitions.

Threshold	PET TBR	tCho/tNAA	Gln/tNAA	Gly/tNAA	Ins/tNAA	tCho/tCr	Gln/tCr	Gly/tCr	Ins/tCr
T/PSEG1.15	1.15	0.22 ± 0.05	0.20 ± 0.09	0.10 ± 0.03	0.60 ± 0.16	0.47 ± 0.09	0.37 ± 0.14	0.22 ± 0.10	1.24 ± 0.29
T/PSEG1.6	1.60	0.31 ± 0.07	0.28 ± 0.13	0.14 ± 0.04	0.83 ± 0.22	0.65 ± 0.12	0.52 ± 0.20	0.31 ± 0.14	1.73 ± 0.41
T/PSEG2.0	2.00	0.39 ± 0.09	0.34 ± 0.16	0.17 ± 0.05	1.04 ± 0.28	0.81 ± 0.16	0.65 ± 0.25	0.65 ± 0.25	2.16 ± 0.51
	Medians and IQRs over all patients								

## Data Availability

Patient-wise evaluation results are available in the provided Appendix A.

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
