# Peer review of "7T HR FID-MRSI Compared to Amino Acid PET: Glutamine and Glycine as Promising Biomarkers in Brain Tumors"

_cancers, 2022, doi:10.3390/cancers14092163_

Round 1
Reviewer 1 Report
Hangel et al. reported a case series of MRSI obtained with 7T MR for glioma patients compared to amino acid PETs. They found glutamine and glycine as promising biomarkers in glioma. MRS and PET are both highly expected for glioma treatment, and this article is very promising. These information can be useful in the clinical decision making for treatments of this type of tumor. However, the manuscript itself has several unclear points and needs to be addressed.
- They included very heterogeneous tumors such as oligodendroglioma and astrocytoma as well as tumor grade from II to IV. It is well known that oligodendroglial tumors act differently for methionine (MET) PET from astrocytic tumors. The mechanism for MET tracer accumulation may be different between oligo and astro. Oligodendrogliomas accumulate MET depending on receptors, while astro does it more depending on intracellular metabolism. Are there any literature information about the difference of oligo and astro in FET PET. Considering this background, it would be better to exclude oligodendroglial tumors, which leads to more clear correlations despite of smaller sample number. The correlation of MRSI and FET PET or MET PET in oligo would be better analyzed independently, although the number is very small.
- What is a biomarker for? If it is for presence diagnosis, in the first place, MET does not accumulate in the low grade astro without Gd enhancement. If it is for malignancy, they should mention to a short summary of FET PET for glioma malignancy for clinical setting in the Introduction section.
- Why they use NAA as a denominator? Although creatine is relatively constant between tumor and normal tissues, NAA is relatively specific for neurons, which would affect both denominator and numerator.
Author Response
Hangel et al. reported a case series of MRSI obtained with 7T MR for glioma patients compared to amino acid PETs. They found glutamine and glycine as promising biomarkers in glioma. MRS and PET are both highly expected for glioma treatment, and this article is very promising. These information can be useful in the clinical decision making for treatments of this type of tumor. However, the manuscript itself has several unclear points and needs to be addressed.
R1.1: They included very heterogeneous tumors such as oligodendroglioma and astrocytoma as well as tumor grade from II to IV. It is well known that oligodendroglial tumors act differently for methionine (MET) PET from astrocytic tumors. The mechanism for MET tracer accumulation may be different between oligo and astro. Oligodendrogliomas accumulate MET depending on receptors, while astro does it more depending on intracellular metabolism. Are there any literature information about the difference of oligo and astro in FET PET. Considering this background, it would be better to exclude oligodendroglial tumors, which leads to more clear correlations despite of smaller sample number. The correlation of MRSI and FET PET or MET PET in oligo would be better analyzed independently, although the number is very small.
Response: You are right that our patient cohort is heterogeneous itself. We also want to point out that 5/6 ODG patients received FET PET. To address your concern, we included a Table for review comparing the cohorts of all patients and patients without ODGs for the TSEG1.6 case. In it, you can see that the exclusion of ODGs would not affect the overall results very much, albeit it would improve some DSC stats slightly.
Stimulated by your query, we added an analysis of separate groups of tumor/classes grades as Supplementary Figure 15 for the TSEG1.6 case. Here, the main observations are that DSCs are overall higher for the high-grade cases compared to low grades, with the exception of ODG 3s, which have the least correspondence (between PET and MRSI but also between PET and T-SEG). Only the Gln ratios to T-SEG perform well in these, which might be interesting to explore with bigger cohorts.
As the overall impact of removing ODGs would be very low, we would stand by including them while adding some of this information to the manuscript.
Regarding your question about literature in ODG/AZ differences in FET-PET, a recent publication (10.1007/s00330-020-07470-9) found FET-PET to be able to discern ODGs from GBMs but not from AZs.
R1.2: What is a biomarker for? If it is for presence diagnosis, in the first place, MET does not accumulate in the low grade astro without Gd enhancement. If it is for malignancy, they should mention to a short summary of FET PET for glioma malignancy for clinical setting in the Introduction section.
Response: We extended the introduction with these aspects.
R1.3: Why they use NAA as a denominator? Although creatine is relatively constant between tumor and normal tissues, NAA is relatively specific for neurons, which would affect both denominator and numerator.
Response: This is of course a valid concern. Still, tCho/tNAA remains the clinical reference standard for tumor evaluation and was employed in the previous studies comparing PET and MRSI, making its use necessary in this context. Therefore, we did both use tNAA and tCr in this study, all tables and supplements include results for both. We had also stated the reliance on metabolic ratios to tNAA and tCr as limitation in that section of the manuscript and hope to overcome their use in the future.
Reviewer 2 Report
Well described review for non radiologist. But I want to add the descriptions abiuyr using machine
Author Response
R2.1: Extensive editing of English language and style required
Response: The manuscript you reviewed was already edited by professional academic editor with AE as native language. Two additional editors have only found miniscule issues with the manuscript. Could you please be specific about the issues you encountered?
R2.1: Well described review for non radiologist. But I want to add the descriptions abiuyr using machine.
Response: Thank you for these kind words. We did not understand your question, but assuming you meant the 7T scanner, we added some more details. Assuming you meant machine learning, no machine learning was used in the evaluation of the study data.
Reviewer 3 Report
This is a highly relevant paper. The study is well done and the paper is written well. The authors refer to their previous study but I would like specification: do the 15 minutes scanning also apply to this data? Is there anything known on the molecular profiles of the tumors except IDH1 status. Gln and Gly levels have been suggested to indicate/ promote cancer stem cell existence. Ideally would be to see if their MRS method is able to detect CSC abundancy. One way is to provide histological data of the corresponding tumors reg. stem cell molecules – i.e. Notch, ZEB1 or CD133 – and correlate to the MRS data. I am sure the used PET imaging can not distinguish CSC high / low- so the proposed method would be not only a support for the clinical imaging, but rather superior. At least discussion this angle in the manuscript would be supportive to indicate the potential of the introduced technology.
Author Response
This is a highly relevant paper. The study is well done and the paper is written well.
Response: Thank you for your kind words!
R3.1: The authors refer to their previous study but I would like specification: do the 15 minutes scanning also apply to this data?
Response: Yes, they do, as detailed in Supplementary Table 1. We clarified this in the manuscript as well.
R3.2: Is there anything known on the molecular profiles of the tumors except IDH1 status.
Response: We extended Table 1 to include more results from clinical molecular pathology.
R3.3: Gln and Gly levels have been suggested to indicate/ promote cancer stem cell existence. Ideally would be to see if their MRS method is able to detect CSC abundancy. One way is to provide histological data of the corresponding tumors reg. stem cell molecules – i.e. Notch, ZEB1 or CD133 – and correlate to the MRS data. I am sure the used PET imaging can not distinguish CSC high / low- so the proposed method would be not only a support for the clinical imaging, but rather superior.
Response: We checked back with our Neuropathology and Neurochemistry, but no such data are available for our patient cohort at this time. It is an excellent idea though that we can hopefully realise in a more advanced study in the future. Especially Glioblastoma MRSI scans over the course of treatment seem of interest.
R3.4: At least discussion this angle in the manuscript would be supportive to indicate the potential of the introduced technology.
Response: We did add this idea to the discussion.
Round 2
Reviewer 1 Report
I'm satisfied with the authors' reply.
Author Response
Thank you!
Reviewer 3 Report
The authors did a moderate effort. At least citing comprehensive papers except cherry picked singular papers regarding the use of GLN GLU as glioma stem cell monitor such as https://pubmed.ncbi.nlm.nih.gov/27389307/ , shall be the basis for more complete interaction with the topic.
Author Response
The authors did a moderate effort.
R3.1 At least citing comprehensive papers except cherry picked singular papers regarding the use of GLN GLU as glioma stem cell monitor such as https://pubmed.ncbi.nlm.nih.gov/27389307/ , shall be the basis for more complete interaction with the topic.
Response: Thank you for your additional feedback. We would like to point out that as none of us authors are in the know about CSC metabolism, we are actually limited to pick the cherries of publications we can find or rely on your comments. We did include your proposed reference and extended the manuscript accordingly.
Thanks again for bringing this interesting connection to our attention.